# Molecular Cloning, Structure and Phylogenetic Analysis of a Hemocyanin Subunit from the Black Sea Crustacean *Eriphia verrucosa* (Crustacea, Malacostraca)

**DOI:** 10.3390/genes12010093

**Published:** 2021-01-13

**Authors:** Elena Todorovska, Martin Ivanov, Mariana Radkova, Alexandar Dolashki, Pavlina Dolashka

**Affiliations:** 1AgroBioInstitute, Agricultural Academy, 8 Dragan Tsankov, 1164 Sofia, Bulgaria; e.g.todorovska.abi@gmail.com (E.T.); m.v.ivanov2010@gmail.com (M.I.); marianaradkova@yahoo.com (M.R.); 2Institute of Organic Chemistry with Centre of Phytochemistry, BAS, Block 9 “Akademik Bonchev” Street, 1113 Sofia, Bulgaria; adolashki@yahoo.com

**Keywords:** *Eriphia verrucosa* (Crustacea; Decapoda; Brachyura), hemocyanin, cDNA cloning, structure, 3-D model, glycosylation, phylogenetic analysis

## Abstract

Hemocyanins are copper-binding proteins that play a crucial role in the physiological processes in crustaceans. In this study, the cDNA encoding hemocyanin subunit 5 from the Black sea crab *Eriphia verrucosa* (EvHc5) was cloned using EST analysis, RT-PCR and rapid amplification of the cDNA ends (RACE) approach. The full-length cDNA of EvHc5 was 2254 bp, consisting of a 5′ and 3′ untranslated regions and an open reading frame of 2022 bp, encoding a protein consisting of 674 amino acid residues. The protein has an N-terminal signal peptide of 14 amino acids as is expected for proteins synthesized in hepatopancreas tubule cells and secreted into the hemolymph. The 3D model showed the presence of three functional domains and six conserved histidine residues that participate in the formation of the copper active site in Domain 2. The EvHc5 is O-glycosylated and the glycan is exposed on the surface of the subunit similar to *Panulirus interruptus*. The phylogenetic analysis has shown its close grouping with γ-type of hemocyanins of other crustacean species belonging to order Decapoda, infraorder Brachyura.

## 1. Introduction

Hemocyanins are blue respiratory proteins in the hemolymph of invertebrate organisms like mollusks (phylum Mollusca) and arthropods (phylum Arthropoda). They play an important physiological role because of their oxygen transportation function within the hemolymph. Hemocyanins, like hemoglobin in vertebrates, are multi-subunit molecules where each subunit (arthropods) or functional unit of a subunit (mollusks) binds to oxygen. Hemocyanins have a binuclear copper active site with two copper ions complexed by six histidine residues. Between the two copper ions, a dioxygen molecule is reversibly bound. This copper type-3 center is also present in other proteins like tyrosinases, catecholoxidases and phenoloxidases, and it is assumed that the hemocyanins evolved from tyrosinase-like ancestral oxygen-binding proteins. Recent studies reviewed by [1] showed, contrary to previous claims that this ancient protein is involved solely in oxygen transport, that hemocyanin and hemocyanin-derived peptides are also linked to key aspects of innate immunity.

Apart from this, Molluscan and arthropod hemocyanins are very large, multimeric, extracellular proteins with a similar active site. Moreover, they easily can dissociate at alkaline pH (e.g., pH 9.6) into functional subunits and reassemble at a near-to-neutral pH (e.g., pH 7.5) into their original quaternary structure. These conformations need the presence of Ca^2+^ and Mg^2+^ ions [2]. On the other hand, the primary, ternary, and quaternary structure of arthropod and molluscan hemocyanin is so different that they are considered as two distinct protein superfamilies. According to DNA sequence and molecular phylogeny analysis, and molecular clock calculations on the separation of molluscan hemocyanin occurred about 740 million years ago [3], and independently, less than 600 million years ago in the case of arthropod hemocyanin [4].

While the structure, function, and role of molluscan Hc are relatively well described, those in phylum Arthropoda are far less known due to the limited number of molecular, biochemical, and microbiological investigations up to date. Hemocyanins have been studied in detail mostly in the subphylums of Chelicerata and Crustacea, although they have been identified in all arthropod subphyla, including Myriapoda [5,6] and Hexapoda [7] as well as in other arthropod taxa like Onychophora and Remipedia [8,9].

The detailed X-ray crystallography and 3D electron microscopy data have shown that the arthropod hemocyanin quaternary structure is based on the hexamer (1 × 6-mer) that contains six heterogeneous subunits of about 75-kDa each (roughly 660 amino acids) and is conserved within all arthropod hemocyanins. Such a structure was observed in lobsters *P. elephas* and *P. interruptus* [10], as well as in other crabs. In arthropod species, hemocyanin hexamers can associate to form higher molecular mass multimers (2-hexamers, 3-hexamers, 6-hexamers and 8-hexamers) [11,12]. In *Scolopendra subspinipes* and *Scolopendra viridicornis* belonging to subphylum Myriapoda, the hemocyanin is composed of 3 × 6 and 6 × 6 subunits, respectively [13] while in the *Limulus polyphemus* (subphylum Helicerata) it is composed of 8 × 6 subunits [14].

The identification of the subunit repertoire and arrangement of hemocyanin can enhance the understanding of hemocyanin evolution and the role of sequence diversity in higher-order oligomer assembly and function, especially in response to environmental changes. The detailed studies on the structure of the arthropodan hemocyanins showed that the monomeric hemocyanin subunit is composed of three distinct domains: N-terminal, M-, and C-terminal domains [15]. The N- and C-terminal domains are more variable in sequence than the M domain and are involved in binding carbohydrates. This characteristic gives to the N- and C-terminal domains potential for immune responses. In the highly conservative M domain, one O_2_ molecule may reversibly bind to two copper ions (Cu+), each of which is coordinated by six histidine residues [15,16]. The C-terminal domain is the most variable across species [17] but also among subunit types with respect to functional diversity, as hemocyanin exhibits also phenoloxidase, anti-viral, anti-bacterial, and antifungal activities in addition to its role in oxygen transport [18,19].

The diversity of hemocyanin functions descends from the evolutionarily derived sequence variances. Three major groups of hemocyanin have been described in the decapod crustaceans: the ancestral and less-understood β-type and the more common α-type and γ-type [20]. The α-type hemocyanin is found in all decapods, while β- and γ-types are absent in many species. Of the α- and γ-types, seven distinct variants have been described in crustaceans [21]. Until now, hemocyanins have been isolated from several organisms belonging to orders Decapoda, Amphipoda, and Isopoda of class Malacostraca [22,23,24,25,26,27,28,29,30,31,32,33,34,35,36].

In this paper, we present for the first time the complete cDNA, and the primary protein sequence of a hemocyanin subunit 5 from a Black sea crustacean, the brachyuran crab *E. verrucosa* and its relationship with other hemocyanins from order Decapoda, class Malacostraca. In addition, analyses of domains, motifs, and residues that are known to be essential for the respiratory proteins and putative glycosylation sites were performed and are discussed herein. The 3D structure analysis of the subunit of EvH was applied for the full characterization of the *E. verrucosa* hemocyanin. Together, these data significantly improve our understanding of the structure, function, and evolution of arthropod hemocyanins. In addition, this information could serve as a base for precise molecular clock evolution in invertebrates and especially in phylum Arthropoda.

## 2. Materials and Methods

### 2.1. Animals

Mature *Eriphia verrucosa* crabs were collected from the Black sea in the south part of the coast near to the Ahtopol village, Bulgaria.

### 2.2. cDNA Cloning and Analyses of Hemocyanin from Eriphia verrucosa

For isolation of total RNA, a hepatopancreas tissue of the crab *E. verrucosa* was used. The tissue was crushed in nitrogen, dispensed into 1.5 mL Eppendorf tubes and immediately homogenized in TRIzol™ Reagent (Thermo Fisher Scientific, Karlsruhe, Germany) in a 1:3 ratio. The isolation procedure was according to the manufacturer’s instructions. The RNA was precipitated from the final solution by adding a one-tenth volume of 3 M Na-acetate and 2.5 volume of ice-cold ethanol, dried, and re-dissolved in RNase-free H_2_O. The quality and integrity of the RNA were checked by measuring the optical densities at 260 and 280 nm, and electrophoretically. cDNA was generated with the RevertAid H Minus First Strand cDNA Synthesis Kit (Thermo Fisher Scientific, Karlsruhe, Germany) using an oligo dT primer according to the manufacturer’s instructions. The Hc cDNA sequences were obtained by PCR using various combinations of oligonucleotide primers (Table 1) generated from the conserved regions of crustacean Hc subunit sequences available in the NCBI database (www.ncbi.nlm.nih.gov) and previously published degenerate primers for the Dungeness crab *Cancer magister* [22]. The Long PCR Taq polymerase (Fermentas) and the AccuPrime™ Taq (Thermo Fisher Scientific, Darmstadt, Germany) polymerases were used for the PCR steps. All amplifications were carried out with PCR buffers containing 1.5 mM MgCl_2_ with the exclusion of the conservative region of Hc, the amplification of which was carried out using 4 mM MgCl_2_ according to [22]. Full-length Hc subunit sequences were obtained by 5′ and 3′ RACE Kit, 2nd Generation (Roche Applied Science, Merck, former Sigma Aldrich), following the manufacturer’s instructions. The 5′ cDNA end was amplified with a nested PCR through the use of the transcript (gene-specific) and an anchor-specific primer. The gene-specific primers are given in Table 1. The PCR products with the expected lengths were ligated into a TA cloning vector pCR2.1 (Invitrogen, Life Technologies), transformed into *E. coli* JM109 and the plasmid DNA of the resulting positive clones were subjected to sequencing. The sequences were obtained by employing a commercial service (Microsynth, Balgach, Switzerland). The partial sequences were assembled by hand with the help of GeneDoc 2.7 [37] and the Vector NTI (Invitrogen Advance 11.5, Thermo Fisher Scientific, Waltham, MA, USA) software. The final cDNA sequence has been deposited at EMBL/GenBank under the accession number KT355032.1. 

The tools provided at the ExPASy server (Swiss Institute of Bioinformatics; http://www.expasy.org/) were used for the identification of the open reading frames (ORFs) and analysis of the amino acid sequences. ExPASy—Compute pI/Mw tool was used to compute (https://web.expasy.org/compute_pi/) the theoretical isoelectric point (pI) and molecular weight of the Hc protein. Signal peptides for transmembrane transport were predicted with SignalP 4.1 [38]. N-Glycosylation sites were predicted with the NetNGlyc 1.0 Server (http://www.cbs.dtu.dk/services/NetNGlyc/).

### 2.3. 3D-Model of E. verrucosa Hemocyanin Subunit

The high degree of sequence identity among arthropodan hemocyanins (>30%), and especially those of the order Decapoda (>60%), suggest a common tertiary structure. X-ray crystallography of Hc from *P. interruptus* and *L. polyphemus* has shown that arthropodan hemocyanins consist of three domains [10,14]. A three-dimensional structure of EvHc5 is presented, using the crystal structure of hemocyanin from *P. interruptus* (California lobsters) as template (PDB code: 1HCY → 1hca.pdb. PDBID (1hcy) was changed to 1HCA (manipulated). Both PDBIDs 1hcb and 1hcd were used temporally at modeling. The proposed 3D model is built using a semi-automatic 3D site modeled on Swiss-Prot, BLAST, ProMod3, RASTOP [39,40]. A number of programs have been implemented sequentially. The graphical user interface MolIDE [41] was used for homologous modeling with SCWRL4 [42], which includes PSI-BLAST searches of PDB, PSIPRED predictions of the secondary structure, structurally assisted alignment editing and modeling of chains and side chains. For homologous modeling of protein structures, an additional Modeller [43] with graphic processing of Chimera [44], DeepView—Swiss-Pdb Viewer [45] and SWISS-MODEL was applied [46]. Based on the results obtained from the programs, a 3D model of the EvHc5 subunit was built.

### 2.4. Phylogenetic Analysis

Multiple alignment of the amino acid sequences that includes the *E. verrucosa* Hc sequence (ALF44611.1) along with other crustacean Hcs with an identity higher than 60%, was constructed. The software MEGA, version 6.06 and Neighbor-Joining method was used with a bootstrap value of 1000.

## 3. Results

### 3.1. cDNA Sequence of E. verrucosa Hemocyanin Subunit

By reverse transcription-PCR using two degenerate primers corresponding to the highly conserved regions of arthropodan Hc, the copper CuA site (CuA I) and the copper CuB site (CuB I), followed by cloning, several clones were obtained and sequenced. Multiple alignment showed the presence of only one Hc subunit those sequence corresponds to the conserved region of the arthropodan Hc.

The missing 5′ and 3′ ends of *E. verrucosa* Hc were obtained by 3′ and 5′ RACE. In the case of 3′ RACE, only one gene-specific primer was used together with the 3′anchor-specific primer, while for the amplification of 5′ end a nested PCR using two gene-specific primers together with the 5′anchor-specific primer was performed. The resulting fragments were with overlapping ends, which allowed the correct assembling of the complete cDNA of *E. verrucosa* Hc of 2254 bp (Acc No KT355032.1). It covers the complete coding region plus the 15 bp 5′ and 201 bp 3′ untranslated regions and 13 bp poly A tail. Within the 3 ‘UTR region, the polyadenylation signal (AATAAA) is located at position 2159–2164 bp (Figure 1).

The open reading frame encompasses 2022 bp and encodes a polypeptide consisting of 674 AK (ALF44611.1). The predicted protein subunit has a mass of 77.1 kDa (77,095.77 Da) and is acidic (isoelectric point pI = 5.32). It has an N-terminal signal peptide of 14 amino acids (Figure 1) as expected for proteins synthesized in hepatopancreas tubule cells and secreted into the hemolymph [47,48,49].

### 3.2. Homology and Phylogenetic Analysis of E. verrucosa Hemocyanin Subunit

Blast searches and pairwise amino acid sequence comparison revealed that the hemocyanin of *E. verrucosa* (Decapoda, suborder Pleocyemata, infraorder Brachyura) displays high sequence identity (>60%) with the 87 hemocyanin sequences from organisms belonging to the three orders of class Malacostraca: Decapoda, Amphipoda, and Isopoda.

In this study, the phylogenetic analysis was performed with hemocyanin sequences isolated from organisms belonging to order Decapoda (sub-orders Dendrobranchiata and Pleocyemata) and ifraorders Brachyura, Achelata, Astacidea, Caridea, and Penaeoidea. The hemocyanins from these species showed an identity between 67.6% and 85.0%, with the *E. verrucosa* hemocyanin subunit. The constructed phylogenetic tree consists of two main clusters (Figure 2). Cluster 1 is composed of four sub-clusters. The first sub-cluster includes 7 hemocyanin sequences isolated from species *M. magister* (subunits 4, 5, and 6), *C. sapidus*, *E. sinensis* (subunit 6), *S. paramamosian* (subunit 1), and *S. serrata* (subunit 1) having a homology with EvHc from 81.7% to 85.0%. The *E. verrucosa* hemocyanin subunit was grouped together with all these hemocyanin sequences within the sub-cluster. The homology analysis using the BLAST algorithm showed that *E. verrucose* amino acid sequence exhibited the highest sequence identity (85.0%) with the subunit 5 of *M. magister*, which gave us the presumption to speculate that the isolated from *E. verrucosa* hemocyanin is subunit 5. The second sub-cluster contains hemocyanin sequences isolated from *C. aestuarii* (subunit 2), *M. magister* (subunit 3), and *P. trituberculatus* (subunit 2), having an identity from 69.7 to 74.1% with EvHc. The third sub-cluster includes two *P. interruptus* hemocyanins with an identity of 67.6% with EvHc. The fourth sub-cluster includes hemocyanins isolated from *M. nipponense* (Mn-Hc1) and *P. vannamei* (Hc subunit 1 and Hc V4). All sequences included in cluster 1 belong to γ-type hemocyanin, which is a clear evidence for belonging *E. verrucosa* Hc subunit 5 to this type of hemocyanins. The second cluster consists of two sub-clusters, which separate α-type hemocyanins isolated from *H. americanus*, *M. nipponense* (MnHc-2) and *P. interruptus* from β-type Hc of *M. magister* (subunits 1 and 2).

### 3.3. Structural Features of the E. verrucosa Hemocyanin Subunit 5

Multiple alignment (Figure 3) of *E. verrucosa* Hc subunit 5 with hemocyanin sequences of known structure like subunits 4, 5, and 6 of *Metacarcinus magister* (AAW57892.1; AAW57893.1; AAA96966.2), *Callinectes sapidus* (AAF64305.1), *Carcinus aestuarii* (P84293.1), *Palinurus interruptus* Hc subunit A (P04254) showed that Domain 1 located between the amino acid residues (1–194), Domain 2 extends from residue 195 to residue 420, and Domain 3 is from 421 till 674 residue (Figure 3). The domain borders were determined according to [24].

The 3D model structure of the EvHc5 showed also the presence of 3 domains (N-terminal or Domain 1; M- or Domain 2 and C-terminal or Domain 3) which correlates with the 3D structure of the subunit type II of *L. polyphemus* and *P. interruptus* (Figure 4a).

Domain 1 is quite variable and is made up primarily of α-helices forming an oxygen access channel to the molecule. It contains a key tyrosine residue at position 45 in EvHc5 (Figure 3) that acts as a gate-keeper to the active site within the central Domain 2. Domain 2 is the most conserved region and comprises the oxygen-binding Cu-A and Cu-B sites each consisting of an antiparallel α-helices pair and three Cu-binding histidine residues (Figure 3 and Figure 4b).

In the EvHc5, the CuA helices pair extends from residue 209 to residue 223 (helix 2.1) and from residue 238 to residue 261 (helix 2.2). The histidines binding to the Cu atoms are located at positions 215 and 219 (helix 2.1) and at position 246 (helix 2.2). The CuB helix pair extends from residue 364 to residue 377 (helix 2.5) and from residue 401 till 419 (helix 2.6). The Cu binding histidines in CuB are located at positions 366 and 370 (helix 2.5) and at position 406 (helix 2.6).

Hemocyanins contain very important residues for their structure and function, some of which are the tryptophan residues (Figure 3). The positions of eleven Trp residues of EvHc5 are shown in Figure 4c, which corresponds to the position of these residues in arthropodan hemocyanins and are exposed on the surface of the molecule. Multiple alignment presented in Figure 3 showed the conservative positions of tryptophan residues in Domain 1 and Domain 2 in all species. The exception was found in Domain 3 of *P. interruptus.* In this species absence of tryptophan residues at position 522 and 531 was observed in comparison with the Hc sequence of *E. verrucosa* and the rest of hemocyanin sequences.

Three phenylalanines that stabilize the binding of oxygen, (96), (242 and 402) were found according to [9]. The phenylalanine in the first domain of the EvHc5 is assumed to be a key residue in the regulation of the oxygen affinity.

Domain 3 is rich in β-sheets and forms a β-barrel structure, and contains a long β-hairpin that reaches towards Domain 1 (Figure 4).

### 3.4. Glycosylation Sites in the E. verrucosa Hemocyanin Subunit 5

Glycosylation is frequently observed in crustacean hemocyanins [27,50,51,52,53].

The native hemocyanin isolated from *E. verrucosa* is glycosylated but the concentration of monosaccharides is very low [54]. To determine the position of potential glycosylated sites of EvHc5, a 3D-model was performed, which showed the glycosylated sites exposed on the surface of the subunit at the same place as in *P. interruptus* (Figure 5). The location of potential glycosylation centers to which carbohydrate chains may be attached on the surface of the molecule are shown in Figure 5.

Sugar analysis of CaeSS2 Hc [27] showed that it is glycosylated, and oligosaccharide chains are connected to three O-glycosylated and one N-glycosylated site. The possible sites for N- and O-glycosylation in EvHc5 were determined by comparison with the CaeSS2 Hc and other closely related crustacea Hc (Figure 3). The CaeSS2 Hc contains one putative N-linkage site at position 309-311 with a consensus sequence Asn-Gly-Ser, typical for N-glycosylation [27,53]. However, at this position, the Asn residue is substituted by Asp in EvHc subunit 5, which means that it could not be N-glycosylated. An absence of N-glycosylation sites in *E. verrucosa* Hc subunit was also confirmed using the NetNGlyc 1.0. Server. Nine putative O-glycosylated sites were identified in EvHc5, which suppose glycosylation at these sites (Figure 3). Seven of them at positions 28–30 (SerThrSer), 170–172 (ThrAsnSer), 184–186 (ThrGlnThr), 195–197 (ThrGlySer), 443–445 (SerPheSer), 546–548 (ThrArgSer), and 575–577 (SerGlySer) are exposed on the surface.

The O-linkage sites at positions 170–172 (ThrAsnSer), 184–186 (ThrGlnThr), 195–197 (ThtGlySer), are conserved in the Hc of *C. aestuarii*, *P. interruptus*, *C. sapidus*, and subunits 4–6 of *M. magister* (Figure 3). The position 399–401 (ThrAlaThr) was observed in hemocyanins from crustacean organisms, while position 546–548 (ThrArgSer) was observed only in EvHc5 and subunits 4–6 of *M. magister*. Similarly, position 575–577 (SerGlySer) was found in EvHc5, *C. sapidus* Hc and in the subunits 4–6 of M. magister (Figure 3).

## 4. Discussion

Oxygen transportation is one of the most important events for living organisms. Among them, mollusks and arthropods have blue blood because they utilize hemocyanin, a type-3 copper-containing protein that freely dissolves in hemolymph for oxygen transportation [2,55,56].

In the last few decades, extensive research on hemocyanins in arthropods and mollusks has been provoked for their role in the evolution of invertebrates as well as their involvement in the major physiological processes like energy storage, osmoregulation, aerobic respiration, the innate immune response, and molting regulation, all related to their heterogeneous protein structure. The studies on sequence diversity of this class of proteins and their subunit composition are mostly related to crustacean adaptation to environmental challenges. In recent years, the research on the structure of Hc in arthropods like in mollusks is also focused on the presence of N- and O-glycans, which are a prerequisite for their usage as therapeutic agents in bacterial, viral, and cancer treatment because many glycans act as receptors for bacteria, viruses, and other pathogens [51,52,53].

With the progress in biotechnology and the development of next-generation sequencing technologies in the last decade, the studies on these important proteins made enormous progress in their isolation, structural, and functional characterization. Until now, the primary structure of hemocyanins have been described in several organisms belonging to suborders: Dendrobranchiata (*L. vannamei*, *L. monodon*, *M. japonicas*, *F. chinensis*, *F. merguiensis*, *M. magister*, *A. vulgare*) and Pleocyemata (*E. sinensis*, *A. moluccensis*, *C. sapidus*, *C. aestuarii*, *C. multidentata*, *C. quadricarinatus*, *P. carinicauda*, *P. interruptus*, *P. vulgaris*, *P. elephas*, *M. nipponense*, *M. rosenbergii*, *S. paramamosain*, *S. serrata*, *H. americanus*, *P. clarkii*) of order Decapoda; suborders Senticaudata (*G. roeselii*) and Corophiida (*C. scammon*) of order Amphypoda and suborders Limnoriidea (*L. quadripunctata*), Cymothoida (*E. pulchra*), Oniscidea (*P. scaber*), and Peracarida (*A. vulgare*) of order Isopoda, all belonging to subphylum Crustacea [22,23,24,25,26,27,28,29,30,31,32,33].

### 4.1. Isolation, Structural, and Functional Properties of E. verrucosa Hemocyanin Subunit 5

To identify homologs of hemocyanin genes in family Eriphidae, infraorder Brachyura we used combined methods (RT-PCR and RACE) based on the sequence similarity with other hemocyanins from class Malacostraca, order Decapoda. The isolated from the Black sea crab *E. verrucosa* cDNA encodes a protein with a length of 674 amino acids, having a putative signal peptide of 14 amino acid at its N-terminal end. The signal peptide ends consist of an Ala-X-Ala motif, which is frequent prior to the cleavage site of the signal peptide. It is usually located at the first 15–30 amino acids [38]. However, often signal sequences do not share sequence similarity in Arthropods and some can be more than 50 amino acid residues long. They are important for trans-membrane transport via endoplasmatic reticulum and are specific characteristics for secretory and membrane proteins.

In our preliminary study, we have identified a high identity (65%) between EvH and CaeSS2 of *C. aestuarii* (P 84293.1). These values are very close to the homology reported between the subunits of other hemocyanins and we have suggested that this fragment belongs to EvH4 or EvHc5 [54]. The isolated here Hc subunit of *E. verrucosa* showed high sequence identity (>60%) with hemocyanins from organisms belonging to the class Malacostraca: Decapoda, Amphipoda, and Isopoda. The highest homology was found between EvHc subunit and *M. magister* subunit 5 (85.0% identity), which allowed us to classify the isolated *E. verrucosa* Hc as subunit 5. The phylogenetic analysis showed specific grouping of EvHc5 with γ-type hemocyanins of arthropods. Three distinct subunit types (α, β, and γ) of hemocyanins have been identified in malacostraca, which diverged more than 450 million years ago [30,57] and assemble into quaternary structures that may even differ within species. In support of this claim is the study [24] which showed that hemocyanin subunits 1 and 2 of *M. magister* belong to β-type, while the subunits 3–6 are of γ-type hemocyanins like those of shrimp, spiny lobster, and another crabs. *M. magister* hemocyanin obviously lacks α-type subunits in contrast to hemocyanins of other decapods, including lobsters and spiny lobsters [58]. The latest are supposed to evolve at a later time compared to α-and β-type subunits [57].

Like other arthropodan hemocyanins, EvHc5 is an oxygen transporter, because it possesses the conserved domains typical for hemocyanin able to bind copper ions—CuA and CuB regions, respectively. Two oxygen atoms are bound between two copper atoms, CuA and CuB, to form a Cu_2_O_2_ cluster. The copper ions are coordinated by six His residues, with CuA coordinated with His215, His219, His246, and CuB coordinated with His366, His370, and His406 (Figure 4b). The six CuA and CuB binding site histidines in the second domain are completely conserved in the hemocyanins. The presence of six histidine residues in the sequence of *E. verrucose* clearly confirms its belonging to the group of the hemocyanines in comparison with nonoxigene pseudohemocyanines (cryptocyanins), in which between one to five specific histidine residues have been reported [24]. The location of these amino acid residues is strictly specific and they are highly conservative in crustaceans [22]. Some additional amino acid residues, like phenylalanine in Domain 1, are conserved across all studied here Hc belonging to the suborder Decapoda. Phe96 in the EvHc subunit corresponds to Phe49 in hemocyanin of the chelicerate *L. polyphemus* that has been thought to play an important role in the regulation of oxygen affinity, along with two additional conserved phenylalanines in Domain 2 [50]. The role of phenylalanine can be also referred as to as allowing the access of larger phenolic substrates into the oxygen-binding pocket, and thus helping the control of phenoloxidase activity of hemocyanin.

X-ray structures are known for various hemocyanins: (1) the deoxy state of the spiny lobster hemocyanin (*P. interruptus*) [59], (2) the oxy and deoxy states of the homohexamer of the horseshoe crab hemocyanin subunit 2 (*L. polyphemus*) [50,60] and (3) the oxy form of the functional unit ”g” of the Octopus hemocyanin and they revealed the active sites of the proteins. Crystallographic investigations of hemocyanins of Panulirus and Limulus show that hemocyanins in arthropods are structurally composed of three domains [50,59]. Domain 1 contains mainly α-helices structures building a super-secondary structure of a stable helical bundle and is very important for the allosteric regulation of oxygen binding and seems to determine the extent of the oligomerization. Domain 2 includes the regions involved in oxygen binding—CuA and CuB. Both regions CuA and CuB are built of antiparallel α-helices. In arthropodan like in molluscan hemocyanins, CuA and CuB are not equivalent and are likely to play a different role in the biological function. The first copper (I) center (CuA) controls the reactivity of the site, while the second copper (II) center (CuB) plays a complex role in controlling the local conformation and electrostatic effects in the oxygenation cycle [24]. The active site is deeply buried within the protein matrix, and a number of hydrophobic residues, including tryptophans, are involved in the active site pocket. It was proposed that such residues may contribute to stabilizing the hydrophobic core of the protein. A number of hydrophobic residues and tryptophans were also found in Domain 2 of EvHc5. Most of the tryptophans located near the active sites are conserved in Hcs from other arthropods like Hc subunits 4, 5, 6 of *M. magister*, subunit 1 of *P. interruptus*, subunit 2 of *C. aestuarii* and *C. sapidus* Hc (Figure 3).

Domain 3 is structurally very different from the others, as it is rich in β-type structures and forms sheets with a barrel shape. It contains a putative calcium-binding site that can play a structural and regulatory role [60,61]. This domain is responsible for recognizing and binding to bacteria or red blood cells, initiating agglutination and hemolysis in *L. vannamei* [62].

### 4.2. Glycosylation of E. verrucosa Hemocyanin Subunit 5 and Possible Role in Immune Response

Likewise to most proteins, hemocyanins are glycoproteins, with large differences in their carbohydrate content and monosaccharide composition, which regulate immunostimulatory properties [1]. The role of glycans on the surface of the molecule, which can interact and serve as ligands, has been proven. Their location and type is related to antiviral, antibacterial and antitumor activity, as presented for other hemocyanins [27,63,64].

It has been suggested that protein glycosylation generally involves the attachment of glycans to serine, threonine, or asparagine residues, with the most abundant type of O-glycosylation being the attachment of GalNAc (O-N-acetylgalactosamine) to serine or threonine residues [65]. Although most of the arthropodan Hc are glycosylated, the glycans represent only about 1–2% of the protein masses of Hcs.

Previous studies [27,51,66] showed one consensus sequence for N-glycosylation and three for O-glycosylation in the CaeSS2. The N-linked site is observed at position 309–311 in CaeSS2 Hc subunit 2 with a consensus sequence Asn-Gly-Ser, typical for N-glycosylation. The sequence comparison of EvHc5 with the CaeSS2 Hc showed that the asparagine residue is substituted by Asp at this position in EvHc and this subunit could not be N-glycosylated. One N-glycosylation site (AsnThrSer) in Domain 3 was observed in hemocyanin subunits 3–6 of *C. magister* [23], one N-linkage site (position 166) was observed in α-type subunits 1, 2, and 3 of *P. elephas* [58], in β-type subunits c and b of *P. interruptus* (position 470) as well as in *Cherax quadricarinatus*. Several putative N- glycosylation sites (Asn-X(Trp)/Ser) were also observed in the primary structure of all seven subunits of *E. californicus* Hc, but they do not pass through Golgi apparatus and no carbohydrate moiety was detected in the native tarantula Hc [67].

In contradiction to N-glycosylation sites in Hc subunits which are mostly species-specific, O-glycosylation is seemingly more conservative in arthropodan Hcs. In our study nine potential O-glycosylation sites in *E. verrucosa* Hc subunits were found, of which seven are exposed on the surface of the protein. Four of them (ThrAsnSer at position 170–172, ThrGlnThr at position 184–186, ThrGlySer at position 195–197 and ThrAlaThr at position 399–401) were observed in Hc subunit 2 of *C. aestuarii* [27] and in other malacostraca Hcs like *C. sapidus*, *P. interruptus* and *M. magister* (SU4—6), which suppose glycosylation at these positions (Figure 3). Among the remaining additional O-glycosylation sites, those at position 546–548 (ThrArgSer) and at position 575–577 (SerGlySer) were found in *M. magister* Hc SU4—6 and in *C. sapidus* Hc, which means that some O-glycosylation sites are species-specific. In confirmation of this are the glycosylation sites at positions 28–30 (SerThrSer) and 443–445 (SerPheSer), which were found only in EvHc5.

Recently, Hex-type of glycans (for glucose, mannose, and galactose) was reported in *L. vannamei* in addition to the mucin-type O-glycans [68]. Among the nine O-glycosylation sites in the small subunit of hemocyanin isolated from *L. vannamei*, those at positions Thr537, Ser539, and Thr542 in the C terminus are of importance for its immunological function [68]. The authors reported a four-fold reduction of bacterial agglutination and a 0.2-fold reduction in the antibacterial activity after replacement of all three positions with alanine. The observed Hex-type of glycan on site Thr-537 is considered to be a new type for arthropods. The last two positions (Ser539 and Thr542) in the small Hc subunit of *L. vannamei* were conservative in *E. verrucosa* (positions Ser548, Thr551) and *C. magister* (SU4-6) but additional studies like site-directed mutagenesis and the use of mutant Hcs in bioassays will show the importance of the reported here O-glycosylation sites on their antimicrobial and/or antiviral activity.

The importance of O-glycosylation of EvHc5 was confirmed by antimicrobial assay of the subunits 1–5 isolated from the native Hc of *E. verrucosa* [52]. The authors reported the potential of subunit 1, 2, and 5 to be used as substitutes of some commonly used antibiotics to which bacterial resistance has been developed like *E. coli* and *B. subtilis* and other bacterial pathogens (*S. epidermidis*, *S. enterica*, *S. aureus*, and *P. aeruginosa*). Similarly, Becker reported that the *Concholepas hemocyanin* (CCH) subunit CCHA, which has both N- and O-linked moieties, is characterized with better antitumor effects in the bladder carcinoma cell line MBT-2 than subunit CCHB, in which O-linked glycans are nearly absent [69]. Functional diversity of hemocyanins related to their glycosylation has been observed [63,64]. It was reported that the glycosylated *R. venosa* hemocyanin FU, RvH2-c, exhibits antiviral activity while the non-glycosylated unit, RvH2-b, showed no antiviral activity [64].

## 5. Conclusions

Isolation of new hemocyanins and the detail characterization of their structure and organization, as well as N- and O-linked moieties, is essential for their future biomedical and biotechnological applications. In this study, we present in detail the molecular characteristics of a newly isolated member of the crustacean hemocyanin family from the Black sea brachyuran crab *Eriphia verrucosa*.

The comparison of EvHc5 with other members of the hemocyanin family shows that the new protein is a typical oxygen-transporter that belongs to γ-type of arthropodan Hc. EvHc5 is O-glycosylated, which is a prerequisite for its active participation in the immune response against various pathogens like bacteria and viruses.

The study provides a new insight into the sequence and protein structure of a respiratory protein from the decapod crustacean family, which could be used as a base for further isolation and molecular characterization of the remaining hemocyanin subunits of *E. verrucosa*.

## Figures and Tables

**Figure 1 genes-12-00093-f001:**
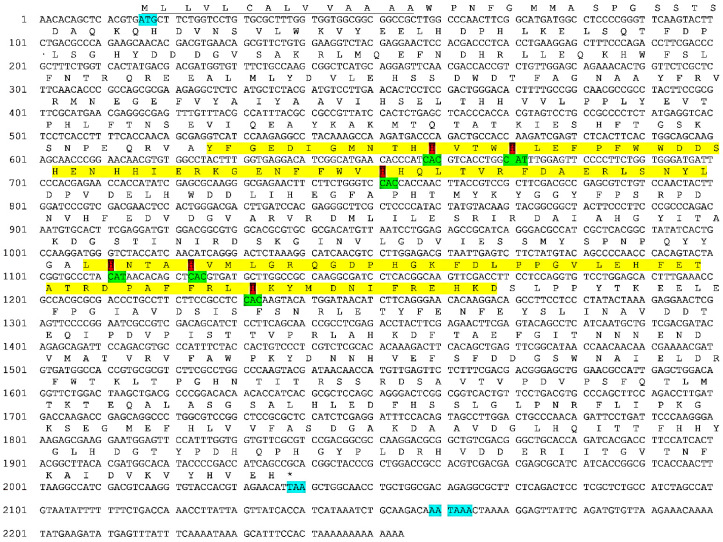
Complete nucleotide sequence, open reading frame (ORF) and the amino acid sequence of *Eriphia verrucosa* Hc. The main elements are highlighted as follows: ATG–start codon, TAA-stop codon and AATAAA–polyadenylation signal with turquoise color, the oxygene–binding regions CuA and CuB–with yellow color. The 6 Cu-binding histidines, among which 3 in CuA and 3 in CuB, respectively are given in red color. The N-terminal signal peptide is underlined.

**Figure 2 genes-12-00093-f002:**
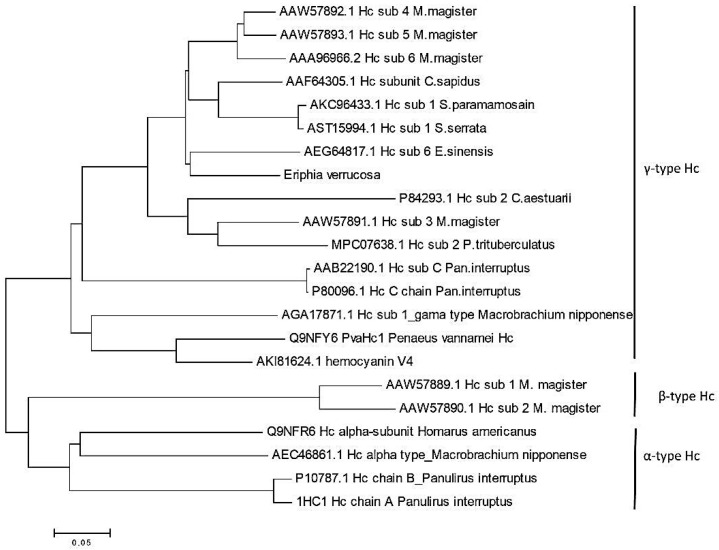
Phylogenetic analysis of *E. verrucosa* hemocyanin. The tree was constructed by MEGA software version 6.06, using the Bootstrap Neighbour-Joining method. The reliability of each branch was tested by 1000 bootstrap replications. The numbers at the nodes represent bootstrap values. Hemocyanin sequences obtained from NCBI correspond to the following species: *Metacarcinus magister*—Dungeness crab (AAW57892.1, AAW57893.1, AAA96966.2, AAW57891.1, AAW57889.1, AAW57890.1); *Eriocheir sinensis*—mitten crab (AEG64817.1); *Calinectes sapidus*—blue crab (AAF64305.1); *Scylla paramamosian* and *Scylla serrata*—green mud crab (AKC96433.1, AST15994.1); *Carcinus aestuarii*—green crab (P84293.1); *Panulirus interruptus*—California spiny lobster (AAB22190, P80096.1). All species belong to order Decapoda, infraorder Brachyura. The crustacean *Portunus trituberculatus*—Gazami crab (MPC07638.1) belongs to order Decapoda, infraorder Achelata; *Homarus americanus*—American lobster (Q9NFR6) belongs to order Decapoda, infraorder Astacidea; *Macrobrachum nipponense*—freshwater shrimp (AGA17871.1, AEC46861.1) belongs to order Decapoda, infraorder Caridea; *Panaeus vannamei* (Q9NFY6; AKI81624.1) belongs to order Decapoda, infraorder Penaeoidea.

**Figure 3 genes-12-00093-f003:**
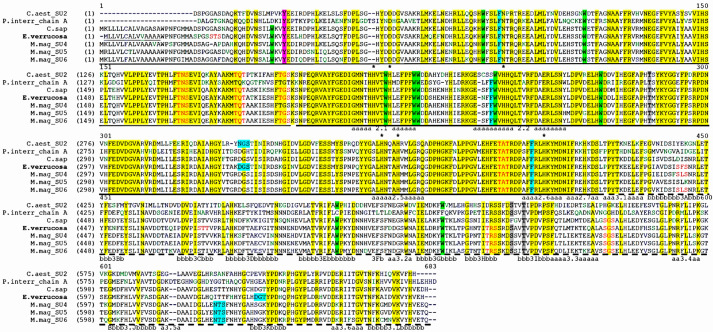
Multiple alignment of EvHc5 with the sequences of the hemocyanin subunits isolated from crustacea species belonging to suborders *Dendrobranchiata* and *Pleocyemata* of the order Decapoda, class Malacostraca. The proposed signal peptide is represented by 14 amino acid (underlined). The conserved residues are highlighted in yellow; the copper-binding histidine residues are shown with asterisks; the conserved phenylalanine (F) with turquoise color; the tryptophan (W) residues with green color; N-glycosylation sites are highlighted in turquoise and the O-glycosylation sites are marked with red letters. The structural elements (α-helices and β-strands) were derived and numbered according to the 3D structure of *P. interruptus* Hc [16]. Domain borders are shown according to [24] as follows: Domain 1—thin black line; Domain 2—thick black line; and Domain 3—dashed line.

**Figure 4 genes-12-00093-f004:**
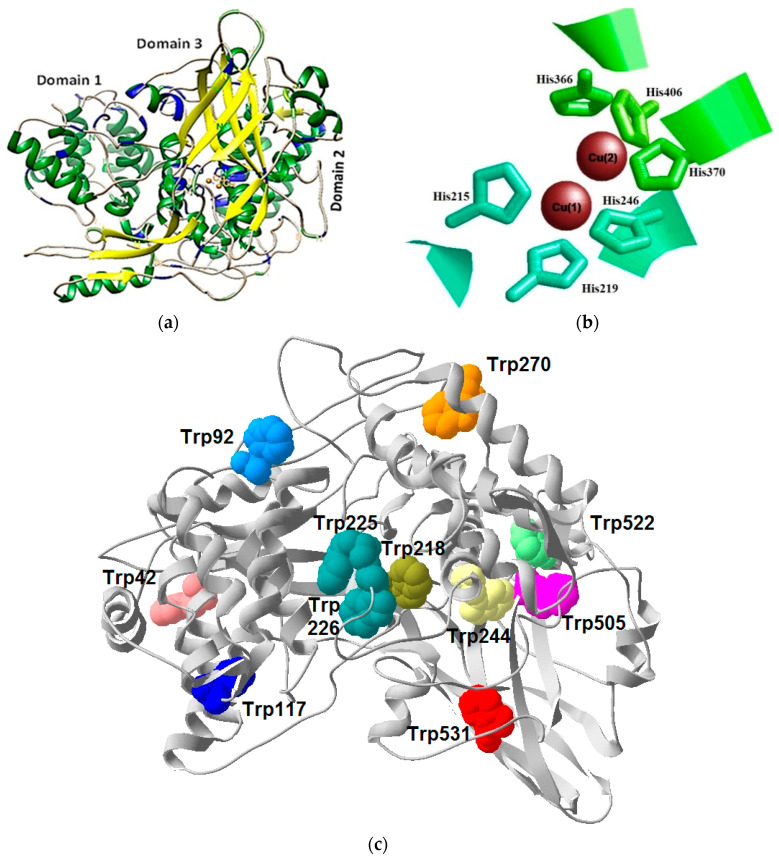
(**a**) Ternary and secondary structure of EvHc5—α-helices (green), β-strands (yellow), random coils (gray). Areas of the chain which carry His residues are colored in blue; (**b**) Active sites of Domain 2 with two Cu ions (brown) and His residues (CuA—215, 219 and 246; CuB—366, 370 and 406; (**c**). Positions of the Trp residues (42, 92, 117, 218, 225, 226, 244, 270, 505, 522, 531) in the EvHc5.

**Figure 5 genes-12-00093-f005:**
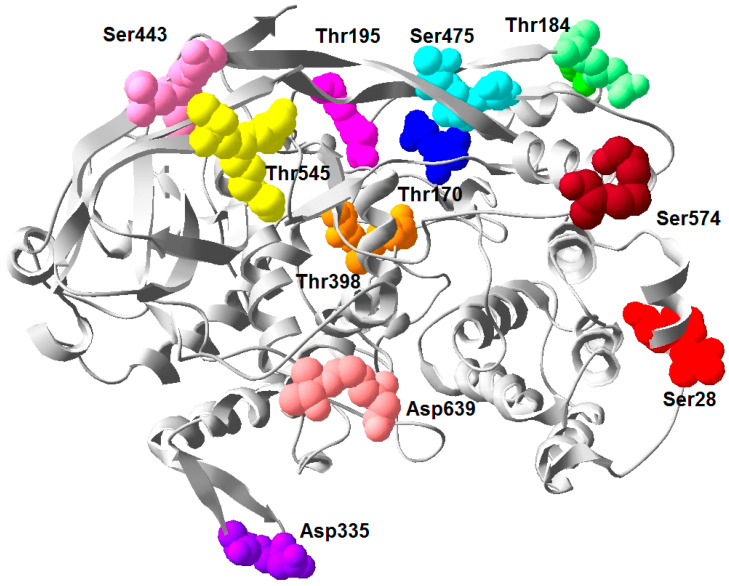
Location of potential glycosylation sites in EvHc5 on positions: 28–30 (SerThrSer), 170–172 (ThrAsnSer), 184–186 (ThrGlnThr), 195–197 (ThrGlySer), 399–401 (ThrAlaThr), 443–445 (SerPheSer), 475–477 (SerThrThr), 546–548 (ThrArgSer) and 575–577 (SerGlySer) in correspondence to the *P. interruptus* glycosylation sites which suppose glycosylation at these sites (Figure 3).

**Table 1 genes-12-00093-t001:** Primers used in PCR amplification of *E. verrucosa* Hc.

Primer	Sequence 5′–3′	References
Degenerate primer F	GAACTTTTTTTTTGGGTTCATCATCAACTTAC	[22]
Degenerate primer R	TGTGTTCTCTGAAGATGTTATCCATGTACTT	[22]
Gene-specific primer F	AAGTACATGGATAACATCTTCAG	In this study
Gene-specific primer R1	TTATGCCGAACTCAGCTGTGAAGT	In this study
Gene-specific primer R2	AAGCATCACGTGAGCTGTGTTA	In this study
3′ anchor-specific primer	GACCACGCGTATCGATGTCGACTTTTTTTTTTTTTTTTV where V = A,C,G	5′ and 3′ RACE Kit, 2nd Generation
5′ anchor-specific primer	GACCACGCGTATCGATGTCGACTTTTTTTTTTTTTTTTV where V = A,C,G	5′ and 3′ RACE Kit, 2nd Generation

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
