# Peer review of "Molecular Cloning, Structure and Phylogenetic Analysis of a Hemocyanin Subunit from the Black Sea Crustacean Eriphia verrucosa (Crustacea, Malacostraca)"

_genes, 2021, doi:10.3390/genes12010093_

Round 1
Reviewer 1 Report
I can see authors have made changes. I am a little confused that original figure 5 and 7 are missing. Why these figures are removed?
Author Response
Reviewer 1 : I can see authors have made changes. I am a little confused that original figure 5 and 7 are missing. Why these figures are removed?
ANSWER : Figures 5 and 7 provide information on the whole hemocyanin molecule, not on the EvH5 subunit, and for more accurate information on the 5 subunit, these figures have been removed.
Figure 5. Orcinol/sulphuric test of: position 1) glucose (1 µg); position 2) glucose (5 µg); position 3) glucose (10 µg); position 4) EvHc (5 µg) and position 5) EvHc (10 µg).
Figure 7. E. verrucosa hemocyanin hexamer structure (1Ñ…6 mer).
Reviewer 2 Report
In this article Todorovska et al. shed new light on the 3D structure and on the phylogenetics of the Hemocyanin subunit 5 from Eriphia verrucosa (EvHc5), a Black Sea crab. Hemocyanins are the oxygen carriers in the hemolymph of invertebrate organisms. Theses hetero-multimeric proteins have two cooper centers that allows them bind oxygen. They also presents different levels of glycosylation that seems to be relevant for the stimulation of the immunitary system as alternative to classical antibiotic treatment. The authors cloned the complete cDNA (2254 bp) corresponding to the EvHc5 protein (674 aa). A phylogenetic study of this EvHc5 is presented and so molecular modeling techniques were use to obtain a 3D model. The domains organization is presented as tryptophan distribution, His-Cu sites and the potential glycosylation sites.
Overall the experiments were well designed and the conclusions valid. However, the article could benefit of reworking in the areas mentioned below (Minor points) in order to improve the article and its results/conclusions. The article needs some kind of homogenization in the text-style. For instance, use as mush as possible EvHc5 instead of EvHc SU5, EvH, EvH FU5, EvH5 and EvHc. A general observation is the massive and irrelevant use of Bold-typo, at least my downloaded copy presents: numbers, letters, sentences, paragraphs … in Bold type.
Specific comments:
Abstract: well presented
Introduction: well presented
Materials and methods: needs some improvements
Results: well presented but needs some corrections
Discussion: well presented
Minor points:
1. Introduction
41-45) Bold typo
70-76) Bold typo
77-78) “diversity” twice in the same sentence, reformulate
79) typo: a-type
83-83) this sentence deserve be joint to the previous paragraph
2. Materials and methods
2.2
99) mL Eppendorf
117) Bold typo
2.3
132-146) Bold typo
132) use “sequence identity” term instead of “sequence similarity” otherwise it is necessary a definition (score) or reference. Generally percentage refer to identity.
135) three domains (to be consistent with the general style)
137) “1HCY.pdb → 1hca.pdb. PDBID (1hcy) was changed to 1HCA (manipulated). Both, PDBIDs 1hcb and 1hcd were used temporally at modeling [39].” information not relevant and confusing. Replace by “as template (PDB code: 1HCY)”.
138) References need it: SwissProt, BLAST …
141-142) References need it: SCWRL4, …
2.4
149) “identity” instead of “similarity”
3. Results
3.1
169) “14 amino acids (fig.4)” wrong figure citation. Fig.3?
3.2
200) g-type
203) b-type
3.3
220-228) Bold typo
226-228) confusing sentence … so the EvHc5 3D model correlate well with its template?
236) b-strands instead of b-sheets
238-247) Bold typo
238) a-helices
238) “oxygen” instead of “O2” (to be consistent with the style)
239) “to enter and exit the molecule”??? needs to be reformulate
239) “key tyrosine residue” (to be consistent with the style)
242) “antiparallel a-helices pair”
243) “helices pair”
245) “helices pair”
Figure 4. a & c) Perhaps the orientation presented in the Magnus 1994 article could help readers. c) Avoid black background and use just numbers for the tryptophan residues in the picture. Legend, prefer 3-letters code for amino acids (1-lettre code it is just relevant for sequence alignments).
251) His residues (CuA – 215, 219 …
252) Trp residues (42, 92 …
255-262) Bold typo
256) tryptophan residues (fig.3).
257) “eleven tryptophan residues” (to be consistent with the style)
263) use: oxygen, (96), (242 and 402) instead of O2, (F96), (F242 and F402)
264) Ertas et al. (2009) (to be consistent with the style)
3.4
271) “with the standard” ? Average? Reference need it here so
271-274) Bold typo
273-274) ¨(Pin HcA 1hca) ¨ just “(fig.5)”
Figure 5. Avoid black background. Perhaps the orientation presented in the Magnus 1994 article could help readers. Or use circles or colors to describe domains boundaries. Legend, prefer 3-lettres code for amino acids (1-lettre code it is just relevant for sequence alignments). ¨(Pin HcA 1hca) ¨
285) (NGS)
289-298) Bold typo
289-292) same information as in legend figure 5, chose one.
289-297) 3-lettres code for amino acids (to be consistent with the style)
4. Discussion
311) some references, it would be very appreciated here
330-331) Bold typo
335-340) Bold typo
338) E. verrucosa
341) (85.0% identity)
343-349) Bold typo
346) b-type
348) a-type subunits
350) a- and b-type subunits
356) (fig.4b) instead of (fig.1)
356) The six CuA (to be consistent with the style)
358) “its belonging” instead of “his belonging”
358) “hemocyanins” instead of “hemocynines¨ ?
361-262) “domain 1” instead of “domain I” (to be consistent with the style)
364) “two” instead of “2”
365) “in domain 2” instead of “in the second domain” (to be consistent with the style)
370) “subunit 2” instead of “subunit II” (to be consistent with the style)
370) “the FU g” ????
373) “alpha sheet” ???
376) oxygen binding
377) “a-spirals”???? … a-helices ?
382-383) some references, it would be very appreciated here
383-387) Bold typo
385) “in Hcs from” instead of “in Hcs in”
388-389) Sentence needs to be reformulated (???)
393) Bold typo
401-403) Please, remove (3-lettres codes)
407) (NGS)
408) Asn asparagine
410) (NTS), 3-lettres code instead
413) 7: “seven” or just remove 7
418) seven
419-426) use 3-lettres codes
429-434) use “Thr537” instead of “Thr-537” format (to be consistent with the style)
455) oxygen-transporter
References
Protein Data Bank … check citation policies at https://www.rcsb.org/pages/policies
My best regards
Author Response
Dear Editor,
I would like to thank the reviewers for the corrections and suggestions made, which are very useful for improving the quality of the article. Text corrections have been made and some of the figures have been changed according to the recommendations made.
Reviewer 2 : In this article Todorovska et al. shed new light on the 3D structure and on the phylogenetics of the Hemocyanin subunit 5 from Eriphia verrucosa (EvHc5), a Black Sea crab. Hemocyanins are the oxygen carriers in the hemolymph of invertebrate organisms. Theses hetero-multimeric proteins have two cooper centers that allows them bind oxygen. They also presents different levels of glycosylation that seems to be relevant for the stimulation of the immunitary system as alternative to classical antibiotic treatment. The authors cloned the complete cDNA (2254 bp) corresponding to the EvHc5 protein (674 aa). A phylogenetic study of this EvHc5 is presented and so molecular modeling techniques were use to obtain a 3D model. The domains organization is presented as tryptophan distribution, His-Cu sites and the potential glycosylation sites.
Overall the experiments were well designed and the conclusions valid. However, the article could benefit of reworking in the areas mentioned below (Minor points) in order to improve the article and its results/conclusions. The article needs some kind of homogenization in the text-style. For instance, use as mush as possible EvHc5 instead of EvHc SU5, EvH, EvH FU5, EvH5 and EvHc. A general observation is the massive and irrelevant use of Bold-typo, at least my downloaded copy presents: numbers, letters, sentences, paragraphs … in Bold type.
ANSWER : EvHc SU5, EvH, EvH FU5, EvH5 are replaced by EvHc5
Specific comments:
- Abstract: well presented
- Introduction: well presented
- Materials and methods: needs some improvements
ANSWER : Additional literatures are included
- Results: well presented but needs some corrections
ANSWER : Several corrections nave been made
- Discussion: well presented
Minor points:
- Introduction
41-45) Bold typo
70-76) Bold typo
77-78) “diversity” twice in the same sentence, reformulate
ANSWER : This sentence is reformulate in The diversity of hemocyanin functions descends from the evolutionarily derived sequence
79) typo: a-type
83-83) this sentence deserve be joint to the previous paragraph
ANSWER : Sentence was deletted
- 2. Materials and methods
2.2
99) mL Eppendorf
117) Bold typo
2.3
132-146) Bold typo
132) use “sequence identity” term instead of “sequence similarity” otherwise it is necessary a definition (score) or reference. Generally percentage refer to identity.
ANSWER : two references are included – 10 and 14
135) three domains (to be consistent with the general style)
137) “1HCY.pdb → 1hca.pdb. PDBID (1hcy) was changed to 1HCA (manipulated). Both, PDBIDs 1hcb and 1hcd were used temporally at modeling [39].” information not relevant and confusing. Replace by “as template (PDB code: 1HCY)”.
138) References need it: SwissProt, BLAST …
141-142) References need it: SCWRL4, …
ANSWER : Six references are included.
- Camacho, C.; Coulouris, G.; Avagyan, V. Ma, N. ; Papadopoulos, J.; Bealer, K.; Madden, T.L. BLAST+:
architecture and applications. BMC Bioinformatics 2009, 10, 421-430 (). DOI: 10.1186/1471-2105-10-421.
- Bernstein, HJ. and Bernstein, FC. RASMOL - Molecular Graphics Visualization Tool, ver.2.7.5.1, (17 July 2009); Philippe Valadon, RASTOP ver.1.3 August 2000. http://www.RasMol.org
- Wang, Q.; Canutescu, A.A.; Dunbrack, Jr R.L. MolIDE 1.7 – Protein 3D Homology Modeling, Nature Protocols, 2008, 3, 1832-1847 https://www.mybiosoftware.com/molide-1-7-protein-3d-homology-model.
- Krivov, G.G.; Shapovalov, M.V.; Dunbrack, R.L.Jr. Improved prediction of protein side-chain conformations with SCWRL4. Proteins 2009, 77(4), 778-795. doi: 10.1002/prot.22488.
- Webb, B.; Sali, A. Comparative Protein Structure Modeling Using MODELLER. Current Protocols in Bioinformatics, 2016, 54, John Wiley&Sons, Inc., 5.6.1-5.6.37. https://www.ncbi.nlm.nih.gov/pmc/ articles/ PMC4186674/
- Pettersen, E.F.; Goddard, T.D.; Huang, C.C.; Couch, G.S.; Greenblatt, D.M.; Meng, E.C.; Ferrin, T.E. UCSF CHIMERA – a visualization system for exploratory research and analysis, J. Comput. Chem. 2004, 25(13), 1605-1612. http://www.rbvi.ucsf.edu/chimera.
2.4
149) “identity” instead of “similarity”
ANSWER : All corrections have been made.
- Results
3.1
169) “14 amino acids (fig.4)” wrong figure citation. Fig.3?
3.2
200) g-type
203) b-type
3.3
220-228) Bold typo
226-228) confusing sentence … so the EvHc5 3D model correlate well with its template?
236) b-strands instead of b-sheets
238-247) Bold typo
238) a-helices
238) “oxygen” instead of “O2” (to be consistent with the style)
239) “to enter and exit the molecule”??? needs to be reformulate - forming an oxygen access channel to the molecule
239) “key tyrosine residue” (to be consistent with the style)
242) “antiparallel a-helices pair”
243) “helices pair”
245) “helices pair”
Figure 4. a & c) Perhaps the orientation presented in the Magnus 1994 article could help readers. c) Avoid black background and use just numbers for the tryptophan residues in the picture. Legend, prefer 3-letters code for amino acids (1-lettre code it is just relevant for sequence alignments).
ANSWER : new figure is presented
241) His residues (CuA – 215, 219 …
252) Trp residues (42, 92 …
255-262) Bold typo
256) tryptophan residues (fig.3).
257) “eleven tryptophan residues” (to be consistent with the style)
263) use: oxygen, (96), (242 and 402) instead of O2, (F96), (F242 and F402)
264) Ertas et al. (2009) (to be consistent with the style)
3.4
271) “with the standard” ? Average? Reference need it here so
271-274) Bold typo
273-274) ¨(Pin HcA 1hca) ¨ just “(fig.5)”
Figure 5. Avoid black background. Perhaps the orientation presented in the Magnus 1994 article could help readers. Or use circles or colors to describe domains boundaries. Legend, prefer 3-lettres code for amino acids (1-lettre code it is just relevant for sequence alignments). ¨(Pin HcA 1hca) ¨
ANSWER : new figure is presented
285) (NGS)
289-298) Bold typo
289-292) same information as in legend figure 5, chose one.
289-297) 3-lettres code for amino acids (to be consistent with the style)
ANSWER : All corrections have been made.
- Discussion
311) some references, it would be very appreciated here
ANSWER : New reference is added. Dolashki, A.; Radkova, M.; Todorovska, E.; Ivanov, M.; Stevanovic, S.; Molin, L.; Traldi, P.; Voelter, W.; Dolashka, P. Structure and Characterization of Eriphia verrucosa Hemocyanin. Mar. Biotechnol. 2015, 17(6), 743-752
330-331) Bold typo
335-340) Bold typo
338) E. verrucosa
341) (85.0% identity)
343-349) Bold typo
346) b-type
348) a-type subunits
350) a- and b-type subunits
356) (fig.4b) instead of (fig.1)
356) The six CuA (to be consistent with the style)
358) “its belonging” instead of “his belonging”
358) “hemocyanins” instead of “hemocynines¨ ?
361-262) “domain 1” instead of “domain I” (to be consistent with the style)
364) “two” instead of “2”
365) “in domain 2” instead of “in the second domain” (to be consistent with the style)
370) “subunit 2” instead of “subunit II” (to be consistent with the style)
370) “the FU g” ????
373) “alpha sheet” ???
376) oxygen binding
377) “a-spirals”???? … a-helices ?
382-383) some references, it would be very appreciated here
ANSWER : New reference is added
383-387) Bold typo
385) “in Hcs from” instead of “in Hcs in”
388-389) Sentence needs to be reformulated
393) Bold typo
401-403) Please, remove (3-lettres codes)
407) (NGS)
408) Asn asparagine
410) (NTS), 3-lettres code instead
413) 7: “seven” or just remove 7
418) seven
419-426) use 3-lettres codes
429-434) use “Thr537” instead of “Thr-537” format (to be consistent with the style)
455) oxygen-transporter
ANSWER : All corrections have been made.
Round 2
Reviewer 1 Report
the revision looks okay now.